# Automatic vs. Human Recognition of Pain Intensity from Facial Expression on the X-ITE Pain Database

**DOI:** 10.3390/s21093273

**Published:** 2021-05-10

**Authors:** Ehsan Othman, Philipp Werner, Frerk Saxen, Ayoub Al-Hamadi, Sascha Gruss, Steffen Walter

**Affiliations:** 1Department of Neuro-Information Technology, Institute for Information Technology and Communications, Otto-von-Guericke University Magdeburg, 39106 Magdeburg, Germany; Philipp.Werner@ovgu.de (P.W.); Frerk.Saxen@ovgu.de (F.S.); Ayoub.Al-Hamadi@ovgu.de (A.A.-H.); 2Department of Medical Psychology, Ulm University, 89081 Ulm, Germany; Sascha.Gruss@uni-ulm.de (S.G.); Steffen.Walter@uni-ulm.de (S.W.)

**Keywords:** pain recognition, facial expression, CNN, multi task learning, random forest

## Abstract

Prior work on automated methods demonstrated that it is possible to recognize pain intensity from frontal faces in videos, while there is an assumption that humans are very adept at this task compared to machines. In this paper, we investigate whether such an assumption is correct by comparing the results achieved by two human observers with the results achieved by a Random Forest classifier (RFc) baseline model (called RFc-BL) and by three proposed automated models. The first proposed model is a Random Forest classifying descriptors of Action Unit (AU) time series; the second is a modified MobileNetV2 CNN classifying face images that combine three points in time; and the third is a custom deep network combining two CNN branches using the same input as for MobileNetV2 plus knowledge of the RFc. We conduct experiments with X-ITE phasic pain database, which comprises videotaped responses to heat and electrical pain stimuli, each of three intensities. Distinguishing these six stimulation types plus no stimulation was the main 7-class classification task for the human observers and automated approaches. Further, we conducted reduced 5-class and 3-class classification experiments, applied Multi-task learning, and a newly suggested sample weighting method. Experimental results show that the pain assessments of the human observers are significantly better than guessing and perform better than the automatic baseline approach (RFc-BL) by about 1%; however, the human performance is quite poor due to the challenge that pain that is ethically allowed to be induced in experimental studies often does not show up in facial reaction. We discovered that downweighting those samples during training improves the performance for all samples. The proposed RFc and two-CNNs models (using the proposed sample weighting) significantly outperformed the human observer by about 6% and 7%, respectively.

## 1. Introduction

Several vulnerable groups, such as infants, intensive care patients, or people suffering from dementia, are not able to self-report their pain [1], and require alternative pain assessment methods to get adequate treatment. Clinical assessment methods such as observing behaviour and physiological signals by human may not always allow for objective and robust measurement for pain diagnosis and do not facilitate continuous monitoring of pain, which is possible with automatic systems [2]. Further, human observer may be influenced by personal factors, such as the relationship to the sufferer [3] and the patient’s attractiveness [4]. In contrast, many machine learning algorithms have been proposed for the automatic recognition of pain intensity, since it has been shown that facial cues are very informative for pain detection [5,6].

At the heart of this research is evidence that the machine is effective compared to human in recognising phasic pain intensity based on analysis of facial expressions in video. Further, the proposed sample weighting method improved the performance of some automated models of pain assessment by downweighting noisy samples. We introduce some automatic pain intensity recognition methods that outperform human observers on the X-ITE database [7], which is a huge database comprising video and other modalities showing reactions of more than 120 healthy adults to experimentally induced painful stimuli. Although this database does not include any person of a vulnerable group, who will benefit most from the pain recognition technology, doing experiments with healthy subjects is an important step for developing the technology. For more details on the dataset see Section 2.1. We propose three methods to predict phasic pain intensity and no pain from facial video: (1) We train a Random Forest classifier (RFc) using a new representation for RGB images (as an automatic baseline approach, called RFc-BL) combining spatio-temporal information. Three images were selected from the second 1, 3 and 4 in each video from X-ITE pain database, converted to gray-scale and resized to 96 × 96, and combined as the red, green, and blue channel of the CNN input image (see Section 2.3.1). (2) We train a Random Forest classifier (RFc) using the Facial Activity Descriptor (FAD) [8] calculated from facial features, namely facial action units (AUs) extracted using OpenFace [9]. (3) Two deep networks were trained: (a) A reduced MobilNetV2 (MNV2) [10] combined with another simple CNN deep learning network, and (b) two combined Convolutional Neural Networks (CNNs).

After observing that misclassified samples often contain low or no facial response to pain (see [11] for details of this phenomenon), we decided to increase the weight of the training samples that show facial response by duplicate training samples with a classification score higher than 0.3. The higher score indicated the samples that contain observable pain reactions, which were successfully predicted in the training set. This weighting method may be counter-intuitive, because it is the opposite of the widely and successfully applied hard-mining strategy. However, many samples do not show facial responses and thus can be considered as noise in the training set. Because manual labels of facial response intensities are not available we suppose that samples that show facial response can be learned and the model should have some confidence in its output. This of course might also downweight some samples that are simply just hard to classify. Nevertheless, the results show that it seems more important to reduce the noise than keeping very hard samples. The test data is kept unmodified in all experiments, that is, samples are never duplicated in the test data. Stratified 5-fold-cross-validation is applied to evaluate the performance of proposed architectures on unseen subjects within all experiments (each fold contains roughly the same proportions of subjects as well as class labels).

The remainder of this paper is organised as follows. Section 1.1 describes the related works to address the relationship between this paper and the reference papers, while Section 1.2 describes the contribution of this work. Section 2 describes the materials and methods for pain intensity recognition using automatic facial expression analysis, including the X-ITE pain database, the input database preprocessing, the proposed Random Forest classifier (RFc) approaches, the proposed deep learning approaches, the multi-task learning, the sample weighting method and the experimental setup. Section 3 presents the experiments result. We discuss and conclude the results in Section 4 and Section 5, respectively.

### 1.1. Related Work

Several studies have shown a significant progress in machine learning architectures that have been used to successfully tackle facial pain expression recognition problem. Facial expressions are an important information channel in non-verbal communication [11,12]. Facial movements express human behaviours [13], emotions [14,15,16,17,18], and pain [17,18,19,20,21]; they are expressed in a similar way by many people of all cultures. The Facial Action Coding System (FACS) has been used to objectively describe facial expressions [4]. The Prkachin and Solomon Pain Intensity (PSPI) score [22,23] and the UNBC-McMaster Shoulder Pain Database [5,22,24] are most commonly used in pain recognition. PSPI can be calculated over each frame in individual images or video to assign pain intensity based on coding the intensity of certain action units (AUs) according to FACS. However, Werner et al. [5,8] recommend to avoid using PSPI due to its several limitations. E.g., in a broader context including basic emotions, which share some AUs with PSPI, many frames showing emotional expressions, e.g., of disgust or happiness, are labelled as painful by mistake. Further, the temporal resolution of PSPI (one pain intensity value per frame) may be misleading when the pain persists for a longer time. E.g., PSPI may go up and down with tension and relaxation of facial muscles although the felt pain is constant or steadily increasing [8]. The PSPI may be zero while some people are actually in pain, because there may be no facial expression present in the lower pain intensities [25,26], which is a general challenge of pain assessment. Further, there are some limitations with UNBC-McMaster, such as the small number of participants (subjects) which may lead to a database bias problem. Movement also varies across subjects during pain stimulation, and their faces are sometimes partially occluded and this represents potential confounds in experimental designs [27]. Gruss et al. [28] reported significant progress in automated pain assessment when using biopotential data. Further, Werner et al. [29] pointed out that head movements and postures might be valuable additional indicators in pain assessment. Many previous works have introduced approaches for facial expression analysis using machine learning models [30]. Several authors designed approaches that were analysed in combination with the pain intensity recognition task. Further, several pain datasets were introduced. We refer the interested reader to the recently published survey article on automated pain assessment methods by Werner et al. [5] for a detailed review of related literature.

Only few prior works in pain recognition have compared the results of automated pain assessment with the assessment of human observers. Bartlett, Littlewort et al. [31,32] showed that an automatic computer vision system can outperform trained human observers in distinguishing real from faked facial expression of pain. Sikka et al. [33] studied pain recognition in a pediatric post-operative context and compared the results of an automated system with the observer pain assessment. The automated recognition was as successful (in ongoing pain) or better (in examination-induced pain) in estimating children’s self report than nurses. In our work, we compare the performance of automated and human pain intensity estimation in an adult population, who experienced pain induced by heat and electricity.

Random Forest classifier (RFc) [34] has been successfully applied in many computer vision works. Werner et al. [35] classified pain intensity using RFc with Facial Activity Descriptor (FAD) [8]. They calculated a face feature vector in the frontal video. Each time series was summarized by several statistics of the time series itself and of its first and second derivative, including mean, maximum, range, time of maximum, and others, yielding a 48-dimensional descriptor per time series. Othman et al. [10] showed how both RFc and reduced MNV2 performed good for binary pain classification (no pain and highest phasic heat pain intensity). The reduced MNV2 architecture used transfer learning with the first 5 inverted residual blocks; which worked well due to the use of less parameters, avoiding the overfitting that easily happens with the full MNV2 on the quite small pain database [10]. In this paper, we utilise these previously mentioned networks and simple convolutional neural network (CNN) to predict phasic pain intensity and no pain from facial video. So this paper advances over [10] by investigating a more complex problem (from binary to multi-class), an advanced recognition approach, and more experiments.

Sharing knowledge among tasks in Multi-Task Learning is often useful when the tasks are similar. It is generally applied by sharing hidden layers while keeping several task-specific output layers. It can help speed up the learning process and reduce overfitting by using shared parameters [36,37]. In this work, we test multitask models but find no improvement in performance compared to the single-task models to recognise pain intensity.

### 1.2. Contribution

This paper, to the best of our knowledge, reports the first comparison between the performance of automatic methods and the performance of human observer for recognising pain intensity on the X-ITE pain database based on facial expression analysis. We have three reasons for comparing machine pain recognition with human performance: (1) Humans are able to recognise and report pain due to their sensitivity to read pain from facial expression [38]; (2) Humans are considered superior when it comes to performing mental tasks [39]; and (3) Success in machine learning applications is often measured by if they are able to reach or beat human performance, therefore many researches evaluate automatic pain recognition performance with human observational assessment [40]. We instructed two observers in this experiment to read motion of facial expression from 10% of the samples in X-ITE pain database and report pain, they labelled only the selected samples. These 10% samples were randomly selected from each class for each subject, thus it ensures each class is represented with the same numbers of samples and avoids sample bias problem.

Another major contribution is the extensive experimental validation of several automated recognition approaches including Random Forest and deep neural network classifiers, multi-task learning, and a newly suggested sample weighting method.

## 2. Materials and Methods

Figure 1 shows an overview of the methodology of automatic facial expression analysis for the faces in video to recognize pain intensity.

We use the Experimentally Induced Thermal and Electrical (X-ITE) Pain Database (see Section 2.1). Within this study, the participants were stimulated with heat and electricity to induce pain in three intensities (low, medium, and high). We focus on analysing the facial expression data for phasic pain intensity recognition. First, we use OpenFace [9] (1) to detect the face from each frame for each participant, and (2) to extract Facial Features (FF). Second, the RGB images were conditioned into the new representation by applying a preprocessing (described in Section 2.3.1) and the FF represented by a Time series Statistics Descriptor [8]. Then, we implemented Random Forests (RF) classifier with RGB images (called RFc-BL, see Section 2.2.1) and with Facial Activity Descriptor (FAD) to recognize pain intensities in X-ITE phasic pain dataset (called RFc, see Section 2.2.2). Further, we introduce two deep learning architectures: (1) The combination of a reduced MobileNetV2 (MNV2) architecture with another simple convolutional neural network (CNN) (also called MNV2 in the following, see Section 2.3.2), and (2) The combination of two simple convolutional neural networks (two-CNNs) (called Two-CNNs or CNNs, see Section 2.3.3). In both, the first branch network gets images from the X-ITE phasic pain database as the input and the second gets output predictions of the RF classifier. We did experiments with multi-task learning, see Section 2.4. We increase the weight of the training samples with more facial response, as described in Section 2.5). In Section 2.6 we give an overview of the conducted experiments.

### 2.1. X-ITE Pain Database

In this section, we give an overview of the X-ITE Pain Database in healthy human participants, which is used to evaluate the performance of different pain intensity recognition methods [7], see Figure 2. In line with Werner et al. [35], we use the same participants (subjects) subset, including samples only, for which data were available from all sensors (frontal RGB camera, audio, ECG, EMG, EDA). The facial expression and head pose are analyzed from RGB video of the face; para-linguistic responses can be analyzed from the recorded audio signal; heart rate and its variability can be analysed from the measured electrocardiogram (ECG); surface electromyography (EMG) has been recorded for measuring the activity of trapezius (neck/shoulder), corrugator supercilii (close to eyebrows), and zygomaticus major (at the cheeks) muscles; electrodermal activity (EDA) has been recorded for measuring sweating. The selected database contains videos of 127 participants aged between 18 and 50 years. Two pain types, heat and electricity were stimulated at the participants in 3 intensities (low, medium, and high) using a thermal stimulator (Medoc PATHWAY Model ATS) and an electrical stimulator (DigitimerDS7A). The intensities used for stimulation were selected individually based on each participant’s pain sensitivity. For this purpose, there was a person-calibration procedure prior to the main stimulation phase, in which the participant self-reported on the pain experienced during several stimuli using the numeric rating scale. In the main phase, each of the 3 pain intensities (times two pain types) was repeated 30 times. For each stimulus, the maximum temperature was held for 5 s, alternated with pauses of 8–12 s, see [7] for further details on the applied protocol to define the participant’s individual pain range and applying painful stimulation depending on the calibrated range. We used these phasic (short) stimulation samples, in total 26,454 samples of frontal face videos of 7 s, which have been cut out from the continuous recording of the main stimulation phase. See [35] for more details about the time windows. Next to these phasic stimuli, longer tonic pain stimuli were applied in the X-ITE study once per intensity for 1 min followed by a pause of five minutes. For more details see [7]. In this work, we focus on analysing the facial expression data involving the phasic pain intensity during the application of the thermal and electrical pain stimuli and no pain.

### 2.2. Random Forest Classifier (RFc)

This subsection introduces an overview of the two proposed Random Forest classifier (RFc) methods that have the same structure with different input types: Random Forest classifier (RFc) baseline method (called RFc-BL method) using a new representation for RGB images, and Random Forest classifier (RFc) method using Facial Activity Descriptor (FAD) of facial action unit time-series.

#### 2.2.1. Random Forest Classifier (RFc) Baseline Method (RFc-BL)

In this work, RFc [34] with 5000 trees was used, which performed the best to predict pain intensity and no pain from samples using automatic facial expressions analysis during the respective video. We applied the RFc with 5000 trees to the images with the RGB format that is introduced in pre-processing subsection (see Section 2.3.1). This is the automatic baseline pain recognition approach and RFc-BL is its abbreviation in the following.

#### 2.2.2. Random Forest Classifier (RFc) with Facial Activity Descriptor (FAD)

We additionally applied the RFc [34] with 5000 trees to facial activity descriptor that is extracted from the corresponding facial features. First, we use OpenFace [9] to extract Facial Features (FF) from each frame for each subject (sample). OpenFace detects the face, facial landmarks, and extracts Action Units (AUs). The FF we use include 17 AU intensity outputs of OpenFace: AU1, AU2, AU4, AU5, AU6, AU7, AU9, AU10, AU12, AU14, AU15, AU17, AU20, AU23, AU25, AU26, and AU45. The FF and their change over time can be represented by a Time series Statistics Descriptor [8,10,41,42]. We obtained a 48-dimensional descriptor per time series. We also applied smoothing for the video time series using the approach suggested by Werner et al. [8], see Figure 3.

### 2.3. The Deep Learning Approaches

This section describes the pre-processsing process and the proposed deep learning architectures that were used to predict the pain intensity from facial video.

#### 2.3.1. Pre-Processing

We applied pre-processing on raw data to obtain the right format of the input to MobileNetV2 (MNV2) architecture. The input image to MNV2 with transfer learning has to be 3 channel (RGB) format, whereas each sample of our data is a video of 7 s length. Figure 4 shows the pre-processing pipeline.

OpenFace is used for face detection and registering face images from each video. Three images were selected from second 1, 3 and 4 for each video based on the study by Werner et al. [11]. They analysed the median flow time series during pain stimulation and baseline, and they showed that the pain activity starts about 2 s after the temperature plateau is reached. The temperature plateau starts at second 1 in X-ITE. The 3 images, which form a spatio-temporal volume, are converted to gray-scale along with resizing them to a common size 96 × 96 (not bigger due to not much difference in the results) and then merged into an RGB image. The gray-scale version of the earliest image in time is used as the red channel, the gray-scale version of the second image is used as the green channel, and the gray-scale version of the last image is used as the blue channel. The resulting RGB image captures a part of the temporal transitions of facial expressions during pain. We use it as the input for the deep neural networks and as the input for the RFc-BL baseline.

Additional input for a second branch network was generated to improve the performance of deep learning models. It consists of 28 × 28 single channel images containing the RFc output predictions of the corresponding sample encoded as numbers 0 to 6. These images are used as the second input of the networks, along with the corresponding image that was generated as illustrated in Figure 4.

#### 2.3.2. Simple Convolutional Neural Network (CNN)

Our own CNN architecture comprises six 2D convolution layers (Conv), each followed by ReLU. The first two Conv have 16 output channels. After each two Conv, we apply a 3 × 3 max-pooling with stride 2 and double the output channels of the next two Conv. The last two Conv have 64 output channels and are followed by global average pooling. The final layer is one dense layer with 1024 neurons that is activated by ReLU. This architecture is used to build the final proposed deep learning architectures, which are presented in the following sections.

#### 2.3.3. Reduced MobileNetV2 (MNV2) with Simple CNN

This section describes how we used transfer learning with a reduced MobileNetV2 (MNV2) [43] to predict pain intensity. We have selected it, as it performed well for binary pain classification from facial video in [10]. The proposed architecture is a combination of two architectures: (1) MNV2 using the first 5 inverted residual blocks after pre-training with ImageNet for the RGB X-ITE images [10], (2) the simple CNN architecture for the RFc prediction images, as shown in the previous subsection. Then, three dense layers were added for combining the concatenated outputs of (1) and (2). The dense layers have 1024, 512 and 128 neurons, respectively and are activated by ReLU. The final dense output layer is activated with the softmax function. The obtained model is trained for 150 epochs with $1\times 10^{-4}$ learning rate.

#### 2.3.4. Two-Convolutional Neural Networks (Two-CNNs)

Our proposed two-CNNs architecture for the pain intensity recognition task is a combination of two simple CNN architectures, one used for the X-ITE images (RGB images) and the other for RFc prediction images (single channel). We combined the two simple CNNs by adding three dense layers that are activated by ReLU, containing 1024, 512 and 128 neurons, respectively. The final dense output layer is activated by the softmax function. Similar to reduced MNV2 with simple CNN, the obtained model is trained for 150 epochs with $1\times 10^{-4}$ learning rate.

### 2.4. Multi-Task Learning in Deep Neural Networks (MTL)

For learning new tasks, we humans often apply the knowledge we have acquired by learning related tasks. This principle is similarly applied in MTL, which has been used successfully across many applications of machine learning and computer vision. In our MTL approach, the deep learning models learn to distinguish pain types, i.e., no pain versus electrical pain versus heat pain, and can be apply this knowledge to another task, which is no pain versus pain intensities. Often one of the sub-tasks can help to improve the performance of other sub-tasks, for example by hard parameter sharing, which is the most commonly used MTL approach in deep learning [44]. It is generally applied by sharing the hidden layers between all tasks, while keeping several task-specific output layers [36]. The more tasks we are learning simultaneously, the more our model has to find a representation that captures all of the task. Thus sharing hard parameter reduces the risk of overfitting [45].

We applied MTL on the proposed deep learning architectures Reduced MobileNetV2 (MNV2) with Simple CNN (MNV2-CNN) and Two-Convolutional Neural Networks (Two-CNNs), however with two output layers instead of one: The first output layer is to recognise baseline (pain intensity 0) versus heat stimuli (H) and electricity stimuli (E). The second output layer is to recognise baseline (pain intensity 0) and pain of the intensities 1, 2, and 3. The two output layers (for two classification tasks) are independently activated with softmax functions. We also experimented with modelling the intensity recognition as a regression task with two alternative versions. In one, the intensity output is activated linearly and trained by optimizing mean squared error (mse) loss. The other alternative uses sigmoid activation and cross-entropy loss. In all cases, the modality classification (first output) uses softmax activation.

### 2.5. Sample Weighting Method

After observing that misclassified samples often contain low or no facial response to pain (see [11] for details of this phenomenon), we decided to increase the weight of the training samples with more facial response. We did this by duplicating some training samples based on the classification score. For the 7-class classification task we trained the two-CNNs model and identified the samples of our training data with prediction scores above 0.3. We duplicated these samples once, which increased the samples size from 26,454 to 34,112 and effectively changed the weighting of the samples in the training. Increasing the weight by duplicating them increased the performance in several experiments. Duplicates give some samples more weight in the training, as some single images could appear multiple times per epoch. Thus, the trained model focuses on different points in the images during training, which causes better results in our experiments, compared to a model trained on the original database. Therefore, the duplicates are desirable because the machine learning model puts more weight on getting these samples (with observable pain recaction) correct and less focuses on samples without an observable pain reaction. Our weighting method is beneficial in our application, because the dataset contains many pain samples without observable facial pain reactions that are impossible to classify correctly based on the facial expression modality (but may be correctly classified using e.g., physiological data). The obtained database was used to train deep learning models, but we excluded the duplicated samples from test sets to ensure comparability of test results.

### 2.6. Experiments

#### 2.6.1. Human Observation Experiment

In order to be able to compare the performance of our recognition models with the performance of human observers, we randomly selected 10% of the X-ITE samples of each of the 7 classes, which were in total 3706. The files of this subset were renamed to hide the ground truth class assignment. Next, the image files were appointed to two human observers with the task to assign one of the 7 classes to each of the samples. The observers received some training on recognising facial expressions of pain. After we investigated the intensity of facial expressions for all samples when expressing pain intensity, we identified the following instructions for observers to label images: no reaction (no-pain), lower reaction (low heat-pain stimuli = H1), low reaction (low electrical-pain stimuli = E1), less moderat reaction (moderate heat-pain stimuli = H2), moderate reaction (moderate electrical-pain stimuli = E2), severe reaction (severe heat-pain stimuli = H3), and more severe reaction (severe electrical-pain stimuli = E3). After the observers completed their class assignment, the labels were compared with ground truth labels. The human observers reached an accuracy of 21.1%. So they outperformed guessing (the Trivial model) by about 7% and outperformed the automatic baseline approach (RFc-BL) by about 1%. The result is far from the desired 100% classification rate, which illustrates how hard this classification problem actually is. Human observers had difficulty distinguishing between no pain versus lower pain due to stimulation, because there often was no facial expression reaction. Pain is expressed differently in individuals. Some persons show no facial reactions for low stimulation, others show even no facial expression during high stimulation. Therefore, human observers and automatic methods cannot accurately distinguish pain intensity in the X-ITE experimental pain dataset, which is in line with findings made on the BioVid Heat Pain Database [11]. To compare the performance of human observation with automatic models, we calculated the accuracy using the automatic predictions of the same samples.

#### 2.6.2. Automatic Pain Recognition Experiments

This section describes the experiments. The experiments were carried out in order to gain insights into pain intensity recognition and to compare the performance of the 3 automatic pain recognition models to the performance of a baseline model and human observation. Further experiments were: Multi-Task learning (MTL) was applied on the proposed deep learning networks, and deep learning models were trained again using the extended databases that were obtained after applying the proposed sample weighting. The accuracy measure is used to measure performance, because it is intuitive and suitable for the data, whose classes were approximately balanced. Stratified 5-fold cross-validation without subject overlap is applied to measure generalization. We also evaluate if the results of proposed models are statistically significantly different compared to human observation results based on the p-values, which were obtained through a paired t-test, with the 7-class classification task.

The next section shows the experiments results of (1) full 7-class classification of phasic pain, (2) 5-class pain intensity recognition, and (3) 3-class pain intensity recognition. We considered the heat and electrical pain (which are two different stimulus modalities associated with different pain qualities), and the four pain intensities (ranging from no pain to high pain). No pain (baseline, pain intensity 0) is denoted with the letter B, electrical pain with E and heat pain with H. The letters E and H are followed by a number between 1 and 3 denoting the stimulus intensity. Table 1 summarizes the notation and the 7 classification tasks that are evaluated in our experiments (one with 7 classes, two with different subsets of 5 classes, and four with different subsets of 3 classes). The reduced MNV2 with the simple CNN is denoted with MNV2, the two simple CNNs architecture is denoted with CNNs, Multi-Task Learning is denoted with MTL, and sample weighting that was applied on deep learning networks is denoted with an appended * (e.g., CNNs*, MNV2*-CNN*).

## 3. Results

### 3.1. 7-Class Pain Intensity Recognition

Table 2 shows the results of classifying all 7 available classes, combining the pain intensity (from 0 = B to 3) and the stimulus modality (heat and electrical pain): B, H1, E1, H2, E2, H3, and E3, see Table 1. All proposed models, before and after using sample weighting (*), perform better than guessing and the automatic baseline approach (RFc-BL). The best performance of the suggested models are given in Figure 5. CNNs* (two simple CNNs architecture with sample weighting) performed the best (27.8%) in phasic pain recognition. Further, the CNNs (without sample weighting) and the MTL-CNN* models perform good with about 27%. Among all the MTL-CNN models, the version with the softmax activated classification tasks performed best. The RFc model also performs well with about 26%. MNV2 and MNV2* perform worse than CNNs and CNNs*, and the automatic baseline RFc (RFc-BL) performs the worst with about 19%.

Table 3 shows the 7-class classification results obtained with the 10% part of the database used for the human observer labelling. Some models statistically significantly outperform the human observer, and their performances are above the human performance by about 6%, see Figure 6. The best performance of our proposed models is 27.8% using CNN*, and the worst is when using RFc-BL model. Whether the entire databases or 10% of the data is used, the performance of most CNNs models increases after applying the sample weighting method (e.g., CNNs* using sample weighting vs. CNNs without using sample weighting (with all data) achieve accuracy means of 25.0% vs. 24.7% and (with 10% data) of 24.8% vs. 24.4%).

### 3.2. 5-Class Pain Intensity Recognition

Due to the results of the 7-class classification, only RFc, CNN and CNN* models were trained with the 5-class classification task. For each pain stimulation modality (heat and electrical), we excluded one of the two lower pain intensities based on their low performance in [35]; the suggested combinations are: B, E1, H1, E3, and H3, as well as B, E2, H2, E3 and H3, see Table 1. The performance of the proposed automatic models for pain recognition with both combinations (see Figure 7 and Table 4) is significantly greater than guessing by about 17% and greater than the automatic baseline approach (RFc-BL) by about 11%, while outperforming RFc-BL by about 8% with the 7-class classification task. On average, RFc and CNNs* models perform similar, 37.6% and 37.7%, respectively.

### 3.3. 3-Class Pain Intensity Recognition

Based on the high performance of the 5-class classification, 3 classes for each stimulation method (heat/electrical) are considered; we excluded H1 or H2 in phasic heat stimuli and E1 or E2 in phasic electrical stimuli, see Table 1. The electrical pain recognition models show better performance than heat pain recognition models for phasic stimulation (see Figure 8 and Table 5 ), which is in line with binary-class classification results in [35]. Further, all obtained results are significantly above the guessing (about 20%) and above RFc-BL (about 11%). RFc and CNNs models give the best results. The average accuracies are 51.3% and 51.7%, respectively.

## 4. Discussion

We conducted several experiments in order to (1) gain insights into recognising pain **intensity with facial expressions in X-ITE database, and (2) to compare the performance of** automatic methods to that of human observer. In line with prior work, the results show that it is possible to recognise no pain versus pain intensities and qualities (heat and electrical stimuli) in acute phasic pain using automatic approaches. According to the obtained results, the recognition of electrical pain stimuli performs better than the recognition of heat pain stimuli using automatic pain recognition. This leads to the hypothesis that electrical pain stimuli may work better due to the earlier and more rapid responses than thermal pain responses. The best obtained result for 7-class classification task is 27.8% when using CNNs* model, and the average accuracy from the best models for 5-class classification task is 37.7%; both significantly above the chance level by 13.6% and 17.8% and above automatic baseline model (RFc-BL) by 8.4% and 11%, respectively. By excluding one of lower phasic pain intensities, pain recognition improved more by about 4% above chance and 3% above RFc-BL. The reason is the difficulty in distinguishing between the lower pain intensity stimulation and baseline, in many cases due to a lack of an observable reaction [11]. By using the proposed sample weighting method (we duplicated the samples with higher classification score), the performance increased within several experiments. MTL has been introduced to improve the performance, however the results are not promising possibly because of the complexity of the problem.

In our experiments, RFc and CNN models performed similar. This may be surprising, because deep learning has outperformed conventional methods in many computer vision and machine learning tasks in the last years. However, in most cases the success of deep learning is based on its ability to benefit from huge datasets. In automated pain recognition, the dataset sizes are quite small, in our case probably too small to learn significantly beyond the domain knowledge used in the RFc method. With larger datasets or better regularization, deep learning may outperform RFc in future studies.

One major goal of automatic pain recognition is to support medical practitioners with assessing pain of patients who cannot utter on their pain experience. The current clinical practice requires a human to observe the behaviour of these patients for assessing their pain [5]. In the future, automatic monitoring systems could provide continuous and objective pain assessments. In this paper, we investigated how well a human observer performs in the 7-class pain intensity and quality assessment task provided by the X-ITE database. We found that the humans reached an accuracy of 21.1%, which is about 7% above chance and 1% above the automatic baseline model (RFc-BL). This quite low number illustrates that it is a hard classification task, even for humans. Possible reasons are discussed in [11] and are mainly related to a lack of observable facial response to pain stimuli. The observers were no caregivers in charge of assessing pain in clinical practice – so this or similar studies should be repeated with such caregivers. However, some evidence in the literature suggests that experience not necessarily improves the performance of humans in judging the pain of others [46].

When comparing with the human performance, we found that some of our automatic models not only performed similar, but significantly outperformed two human observers. The best of our proposed models (CNN*) reached an accuracy of 27.8%, which is about 7% above the performance of two humans. Thus, we believe that the pain expression recognition technology is ready for clinical trials comparing automated tools with clinically established pain assessment.

## 5. Conclusions

As we have seen in the quantitative results in discussion section, automatic pain recognitions models and human observers often confuse no pain and low, moderate, and severe pain of both pain modalities (heat and electrical pain). Figure 9 shows some wrongly classified samples. We investigated the facial expression of all samples, and found plenty of labels (which represent the pain stimulus) that do match the observed facial pain expression due to individual differences in pain sensitivity and expressiveness.

As we have seen in the quantitative results in this section, Some subjects show a lack of facial responses to pain: some have low pain sensitivity resulting in a high tolerance threshold requiring a temperature cutoff to avoid burns; others show low tolerance threshold intentionally or unintentionally during stimulus calibration, possibly because they do not want to feel severe pain. Such inconsistencies between the label and the video may be considered outliers or label noise. Data cleaning by removing such outlier samples may be used for improving the facial expression-based recognition performance. However, this may remove some samples, which are useful for improving multimodal pain recognition because there may be pain responses in other modalities like EMG or EDA although no facial responses shows up. Thus, the proposed sample weighting method could be useful for improving the performance of the multimodal pain assessment by downweighting noisy samples rather than eliminating them as in the cleaning up strategy. In addition, for future data acquisition, advanced study designs and more reliable pain ground truth would be beneficial.

There are some risk factors of using this technology: (1) The database (X-ITE) does not contain any of a vulnerable group, we believe that this system can help to predict pain particularly in vulnerable patients but have not yet implemented it to them; (2) Kunz and Lautenbacher [47] and Rash et al. [48] report how training observers to recognize the various faces of pain improves their ability to accurately recognize pain, but we are certain that machine performance is superior to humans on a wide variety of visual recognition tasks; our unproven hypothesis is increasing training might improve the observers’ ability to recognize pain but not significantly compared with automatic pain recognition; (3) due to the COVID-19 pandemic, it has been difficult to find observers from the healthcare domain; we trained only two observers in this experiment to read motion of facial expression from 10% of the samples in the X-ITE pain database and report pain; the observers were not caregivers in charge of assessing pain in clinical practice; however, some evidence in the literature suggests that experience does not necessarily improve the performance of humans in judging the pain of others [46]. 

## Figures and Tables

**Figure 1 sensors-21-03273-f001:**
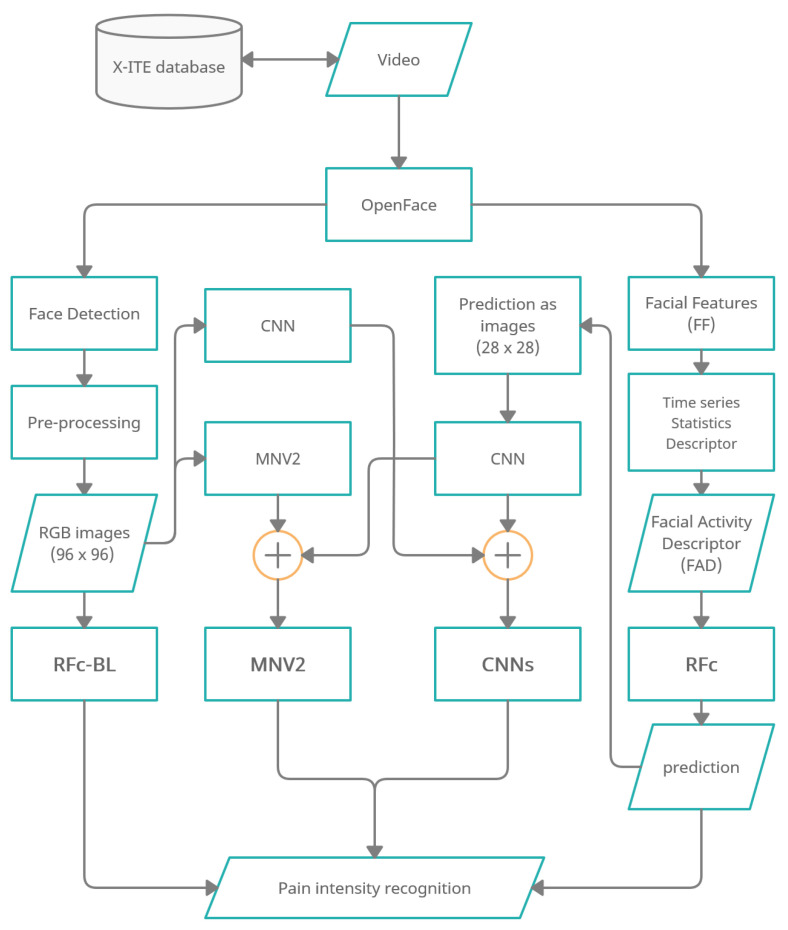
Overview of pain recognition system variants “RFc-BL”, “MNV2”,“CNNs” and “RFc”.

**Figure 2 sensors-21-03273-f002:**
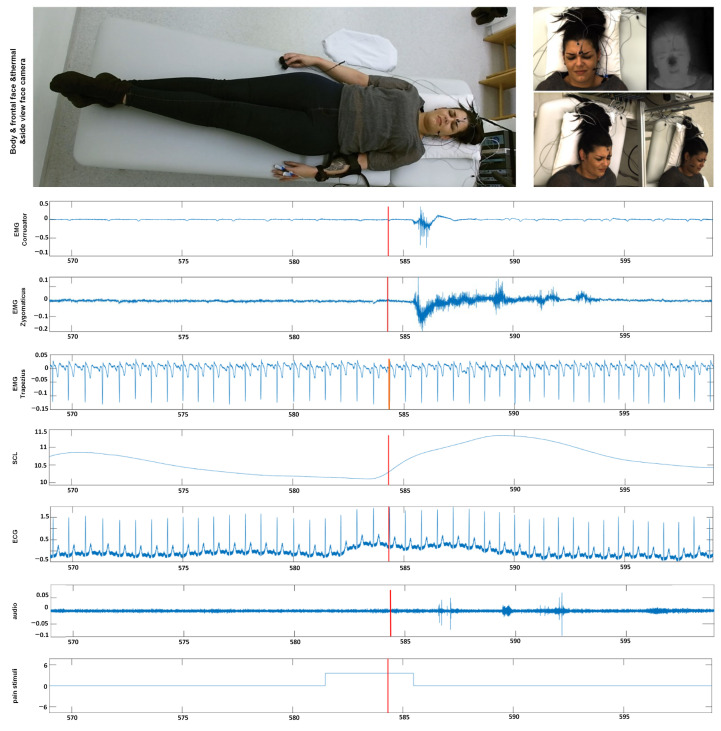
Example data from our X-ITE Pain Database. The representative screenshots of the video signals indicate when one of an intense pain stimulant was applied. The representative screenshots of the video signals (top) show the reactions when one of an intense pain stimuli was applied. The figure depicts plots of the recorded signals (middle part) before, during, and after the application of a pain stimulus (bottom plot). (EMG = Electromyography, SCL = Skin Conductance Level, ECG = Electrocardiogram, M. = Musculus, s = seconds).

**Figure 3 sensors-21-03273-f003:**
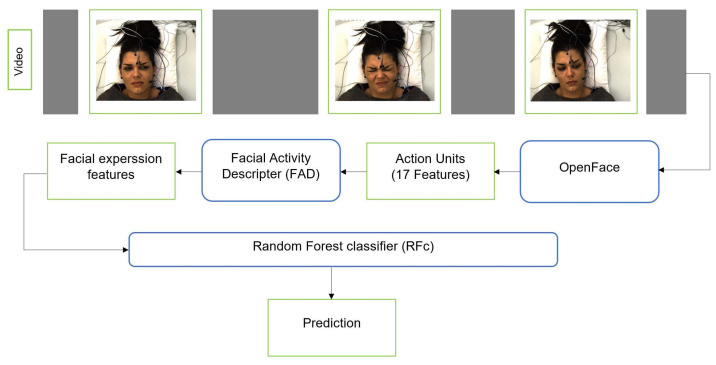
The pipeline of pain classification using RFc with FAD.

**Figure 4 sensors-21-03273-f004:**
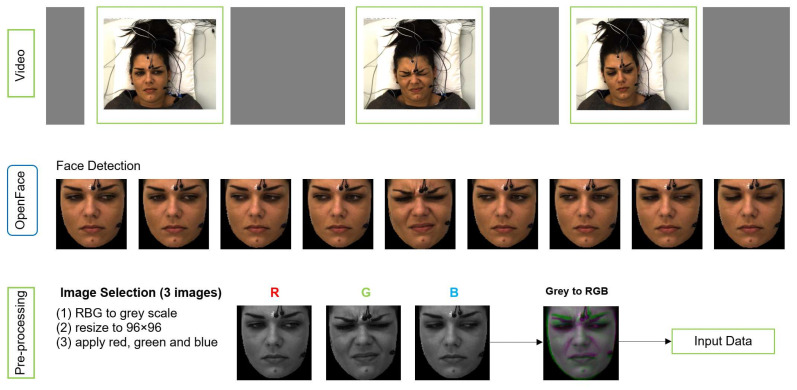
The pipeline of data preparation (pre-processing).

**Figure 5 sensors-21-03273-f005:**
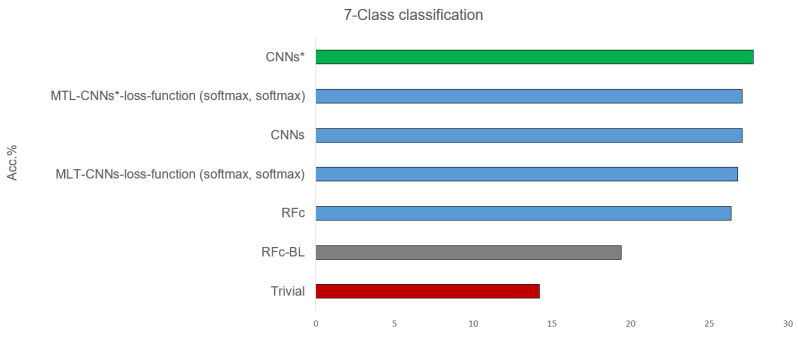
Best recognition models for 7-class classification task compared to baseline and trivial model using all dataset.

**Figure 6 sensors-21-03273-f006:**
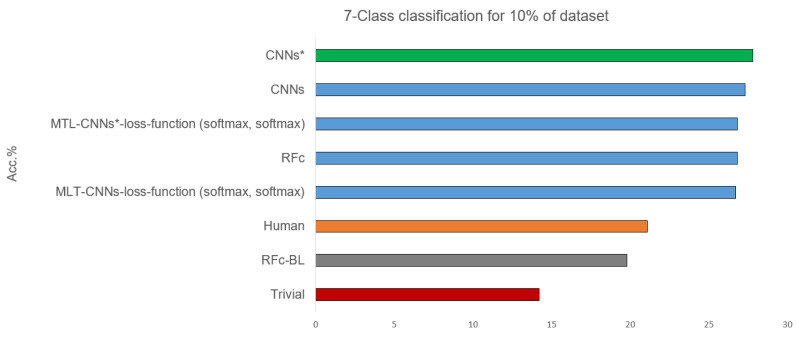
Best recognition models for 7-class classification task compared to baseline and trivial model and human observation using 10% of dataset.

**Figure 7 sensors-21-03273-f007:**
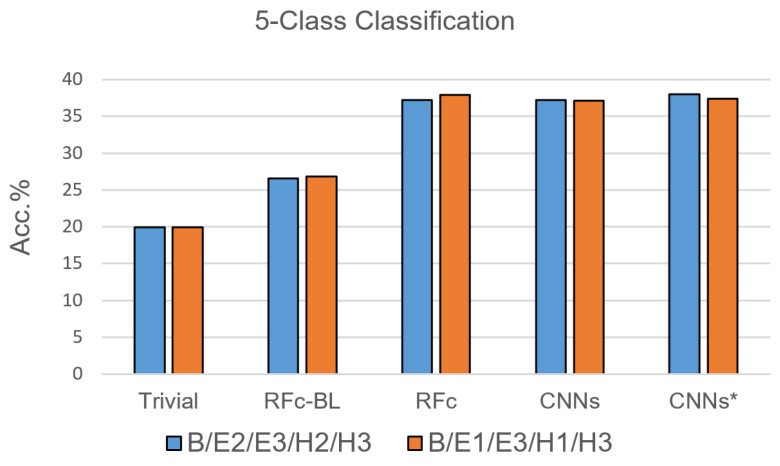
Recognition models for 5-class classification tasks compared to trivial and baseline models.

**Figure 8 sensors-21-03273-f008:**
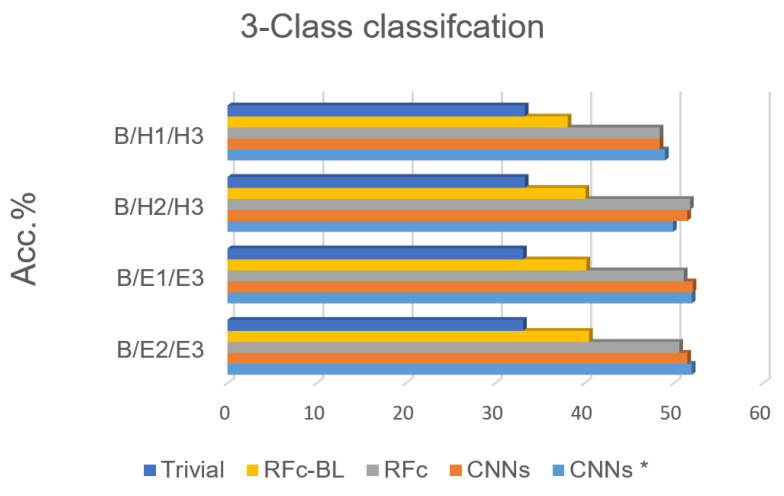
Recognition models for 3-class classification tasks compared to trivial and baseline models.

**Figure 9 sensors-21-03273-f009:**
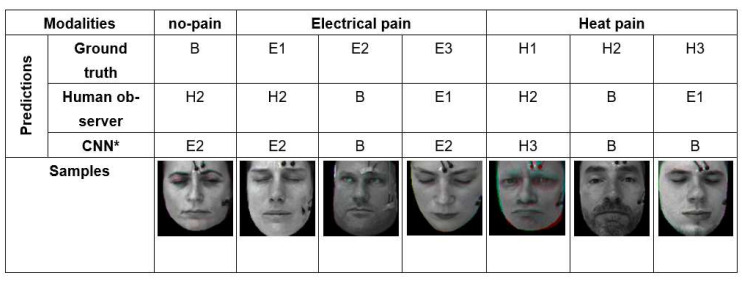
Some examples that the best model (CNN*) and human observer failed in classifying pain intensity and modality due to the difficulty of the problem.

**Table 1 sensors-21-03273-t001:** List of the seven classification tasks used in the experiments (rows). Different combinations of no pain (Baseline = B) versus of 3 pain stimulus intensities and 2 pain stimulus modalities (Electrical = E and Heat = H) evaluated.

Pain Stimulus					
**n-Class**	**Modalities**	**Intensities**			
		**Severe**	**Moderate**	**Low**	**No Pain**
7-class	E	E3	E2	E1	B
	H	H3	H2	H1	
5-class	E	E3	-	E1	B
	H	H3	-	H1	
5-class	E	E3	E2	-	B
	H	H3	H2	-	
3-class	E	E3	-	E1	B
3-class	H	H3	-	H1	B
3-class	E	E3	E2	-	B
3-class	H	H3	H2	-	B

**Table 2 sensors-21-03273-t002:** Recognition approaches accuracy results of phasic stimuli 7-class classification task, the meaning of abbreviations is provided by the list in abbreviations section.

Recognition Approaches	Acc.%	Mean%
Trivial	14.2	-
RFc-BL	19.4	-
MNV2	22.8	**21.2**
MTL-MNV2-loss-function (softmax, softmax)	20.9	
MTL-MNV2-loss-function (softmax, mse)	19.6	
MTL-MNV2-loss-function (softmax, sigmoid)	21.4	
MNV2*	20.4	**20.8**
MTL-MNV2-loss-function (softmax, softmax)	21.7	
MTL-MNV2*-loss-function (softmax, mse)	19.5	
MTL-MNV2*-loss-function (softmax, sigmoid)	21.5	
**RFc**	**26.4**	-
**CNNs**	**27.1**	**24.7**
**MLT-CNNs-loss-function (softmax, softmax)**	**26.8**	
MTL-CNNs-loss-function (softmax, mse)	22.2	
MTL-CNNs-loss-function (softmax, sigmoid)	22.5	
**CNNs***	**27.8**	**25.0**
**MTL-CNNs*-loss-function (softmax, softmax)**	**27.1**	
MTL-CNNs*-loss-function (softmax, mse)	23.7	
MTL-CNNs*-loss-function (softmax, sigmoid)	21.5	

**Table 3 sensors-21-03273-t003:** Recognition approaches accuracy results of phasic stimuli 7-class classification task for 10% of dataset, the meaning of abbreviations is provided by the list in abbreviations section.

Recognition Approaches	Acc.%	Mean%	*p*-Value
Trivial	14.2	-	-
RFc-BL	19.8	-	0.143
MNV2	20.8	**20.6**	0.842
MTL-MNV2-loss-function (softmax, softmax)	20.6		0.645
MTL-MNV2-loss-function (softmax, mse)	19.6		0.133
MTL-MNV2-loss-function (softmax, sigmoid)	21.4		0.636
MNV2*	20.4	**20.8**	0.46
MTL-MNV2*-loss-function (softmax, softmax)	21.5		0.456
MTL-MNV2*-loss-function (softmax, mse)	19.8		0.291
MTL-MNV2*-loss-function (softmax, sigmoid)	21.6		0.164
**RFc**	**26.8**	-	**$1.1\times 10^{-10}$**
**CNNs**	**27.3**	**24.4**	**$2.9\times 10^{-10}$**
**MLT-CNNs-loss-function (softmax, softmax)**	**26.7**		**$9.2\times 10^{-10}$**
MTL-CNNs-loss-function (softmax, mse)	21.8		0.24
MTL-CNNs-loss-function (softmax, sigmoid)	21.7		0.308
**CNNs***	**27.8**	**24.8**	**$7.9\times 10^{-12}$**
**MTL-CNNs*-loss-function (softmax, softmax)**	**26.8**		**$1.8\times 10^{-9}$**
MTL-CNNs*-loss-function (softmax, mse)	22.8		0.015
MTL-CNNs*-loss-function (softmax, sigmoid)	21.6		0.164
**Human**	**21.1**	-	-

**Table 4 sensors-21-03273-t004:** Recognition approaches accuracy results of phasic stimuli 5-class classification task, the meaning of abbreviations is provided by the list in abbreviations section.

Acc.%	B/E2/E3/H2/H3	B/E1/E3/H1/H3	Mean
Trivial	19.9	19.9	19.9
RFc-BL	26.6	26.8	26.7
**RFc**	37.2	**37.9**	**37.6**
CNNs	37.2	37.1	37.2
**CNNs***	**38.0**	37.4	**37.7**

**Table 5 sensors-21-03273-t005:** Recognition approaches accuracy results of phasic stimuli 3-class classification task, the meaning of abbreviations is provided by the list in abbreviations section.

Acc.%	B/ E2/E3	B/ E1/E3	B/ H2/H3	B/ H1/H3	Mean
Trivial	33.1	33.1	33.3	33.3	33.2
RFc-BL	40.5	40.2	40.1	38.1	39.7
**RFc**	50.6	51.1	**51.8**	48.4	51.3
CNNs	51.5	52.1	51.5	48.4	**50.9**
**CNNs***	**52.0**	**52.0**	49.9	**49.0**	51.7

## Data Availability

Data are available from the authors upon request (to Sascha Gruss or Steffen Walter) for researchers of academic institutes who meet the criteria for access to the confidential data.

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
