# Peer review of "Automatic vs. Human Recognition of Pain Intensity from Facial Expression on the X-ITE Pain Database"

_sensors, 2021, doi:10.3390/s21093273_

Round 1
Reviewer 1 Report
- The difference between RFc and RFc-BL should be revealed more explicitly.
- It is necessary to explain how a gray-scale image is merged into an RGB image in more detail.
- Something should be in place of the question mark in [13? –15] (Page 2).
- In the caption of Table 4, the 5-class classification task (instead of 7) should be mentioned. There is a similar remark for the caption of Table 5. Here, the results of the 3-class classification task are presented.
- The performed experiments show that the differences between the results of the random forest classifier and deep learning models are not essential. It would make sense to explain this phenomenon, as the theory says that deep learning should outperform conventional methods.
Reviewer 2 Report
In this study, authors presented automatic pain intensity recognition methods that outperform human observers on the X-ITE database. Overall, manuscript is well-written, however below are my recommendations to further improve it:
- The Introduction section should be modified. It is more like a related work section. Introduction should also include key contributions of the work.
- Related word/Back ground study section should be separate. I would suggest creating separate section of related work and at the end of this section authors should mention key distinguishing features of the study from previous studies as well.
- The methodology section is needed. A brief paragraph is required that explains the methodology and also schematic flowchart should be included to elaborate it.
- Conclusion section should be separate section.
- Minor comment: Reference is missing in Line 37.
Round 2
Reviewer 2 Report
The authors have adequately improved the manuscript. I do not have any further comments.
Author Response
Thank you for your feedback